# Astrocytes in Amyloid Generation and Alcohol Metabolism: Implications of Alcohol Use in Neurological Disorder(s)

**DOI:** 10.3390/cells13141173

**Published:** 2024-07-10

**Authors:** Mohit Kumar, Natalie Swanson, Sudipta Ray, Shilpa Buch, Viswanathan Saraswathi, Susmita Sil

**Affiliations:** 1Department of Pharmacology and Experimental Neuroscience, University of Nebraska Medical Center, Omaha, NE 68198, USA; 2VA Nebraska-Western Iowa Health Care System, Omaha, NE 68105, USA; 3Department of Internal Medicine, Division of Diabetes, Endocrinology, and Metabolism, University of Nebraska Medical Center, Omaha, NE 68198, USA

**Keywords:** alcohol, amyloids, lncRNA BACE1-AS

## Abstract

As per the National Survey on Drug Use and Health, 10.5% of Americans aged 12 years and older are suffering from alcohol use disorder, with a wide range of neurological disorders. Alcohol-mediated neurological disorders can be linked to Alzheimer’s-like pathology, which has not been well studied. We hypothesize that alcohol exposure can induce astrocytic amyloidosis, which can be corroborated by the neurological disorders observed in alcohol use disorder. In this study, we demonstrated that the exposure of astrocytes to ethanol resulted in an increase in Alzheimer’s disease markers—the amyloid precursor protein, Aβ1-42, and the β-site-cleaving enzyme; an oxidative stress marker—4HNE; proinflammatory cytokines—TNF-α, IL1β, and IL6; lncRNA BACE1-AS; and alcohol-metabolizing enzymes—alcohol dehydrogenase, aldehyde dehydrogenase-2, and cytochrome P450 2E1. A gene-silencing approach confirmed the regulatory role of lncRNA BACE1-AS in amyloid generation, alcohol metabolism, and neuroinflammation. This report is the first to suggest the involvement of lncRNA BACE1-AS in alcohol-induced astrocytic amyloid generation and alcohol metabolism. These findings will aid in developing therapies targeting astrocyte-mediated neurological disorders and cognitive deficits in alcohol users.

## 1. Introduction

Dementia is a neurocognitive disorder that involves deficits in memory, cognitive functioning, and executive functioning, resulting in substantial impairment in an individual’s capacity to perform routine activities [1]. The World Health Organization (WHO) has designated dementia as a major health concern. Current data suggest that dementia affects approximately 50 million individuals worldwide, with 10 million being diagnosed annually [2]. The available evidence substantiates the involvement of substance use, including opioids, methamphetamine, and alcohol, in the manifestation of cognitive decline and may potentially augment the vulnerability to the initiation of dementia [2,3,4]. Alcohol is often identified as a potential modifiable determinant in the risk assessment for dementia [5]. In the United States, alcohol stands as one of the predominant substances consumed with the potential for misuse. In 2022, data indicated that 84.1% of individuals acknowledged having consumed alcohol at least once in their lifetime, and a significant subset, comprising 14.5 million individuals aged 12 years and above, were diagnosed with alcohol use disorder (AUD) (NIAAA, 2023). Numerous epidemiological studies have identified a link between low to moderate alcohol consumption and a heightened risk of cognitive decline and dementia [6]. A cohort study involving 42,870 Japanese individuals in the age range of 54–84 years revealed that consistent weekly intake of ≥450 g of alcohol from midlife was associated with an elevated risk of disabling dementia [7].

Alcoholism can elicit multifaceted impacts on neural functions and behavioral patterns, with various determinants shaping these manifestations. Prior research employing diverse imaging techniques has demonstrated a global diminution in brain volume and a consistent correlation between excessive alcohol intake and brain damage [8]. Recent research indicates that males exhibiting alcohol dependence present reduced volumetric measurements in the amygdala and hippocampus, specifically within its subiculum subfield and hippocampal CA1 region, compared to control subjects [9]. Additionally, prolonged alcohol intake has been observed to alter dopamine secretion in a sex-specific manner and influence its regulatory feedback mechanisms within the caudate and putamen of the dorsal striatum, affecting synaptic plasticity [10]. Furthermore, meta-analysis investigations have indicated a correlation between adolescent binge drinking and general cognitive deficits, particularly in decision-making and inhibitory control [11]. Several investigations have shown a dose-responsive association between alcohol consumption and cognitive deficits [12,13,14]. An increased frequency of alcohol consumption over a span of four years was linked to a decline in verbal memory and visuospatial skills. Additionally, elevated estimated peak blood alcohol concentration (BAC) in rapid-binge (10+ drinks per occasion) drinkers over six years was correlated with deteriorations in verbal learning as well as immediate, short-term, and long-term delayed and cued memory recall [11,15,16,17].

In Alzheimer’s disease (AD), cognitive decline is thought to result from an early disruption and build-up of amyloid β (Aβ), often occurring before dementia manifests [18]. The generation of Aβ involves the sequential action of β-secretase (also known as beta-site APP cleavage enzyme, BACE) and γ-secretase enzymes on the amyloid precursor protein (APP). This process plays a vital role in AD pathogenesis [19]. Alterations in the activity of BACE1 and/or γ-secretase enzymes culminate in the generation and aggregation of neurotoxic Aβ, subsequently influencing the development of AD pathogenesis [20]. BACE1 is increasingly recognized as a central modulator of amyloidopathy [21]. Recent evidence, particularly from studies using APP/PS1 AD transgenic mouse models, suggests that long-term exposure (five weeks) to high doses (13–18 g/kg) of alcohol can induce cognitive impairments and enhance the enzymatic activity of BACE1 and γ-secretase involved in APP processing. This upregulated enzymatic activity is associated with increased accumulation and deposition of Aβ peptides, potentially leading to the progression of AD [22]. Contemporary research has spotlighted pivotal factors that influence BACE1 production. Notably, long noncoding RNAs (lncRNAs) have emerged as significant contributors in this domain [23]. Specifically, lncRNA BACE1-antisense (BACE1-AS) has been identified to enhance both the stability of BACE1 mRNA and its protein presence in neuronal cells, culminating in augmented Aβ production in individuals diagnosed with AD [24]. Alcohol intake has been posited as a potential risk factor for AD [24] and alcoholic dementia [25]. A cohort study involving 360 early-stage AD patients from Baltimore, Boston, Paris, and New York demonstrated that heavy alcohol intake correlated with an accelerated rate of cognitive decline, implying a potential acceleration in AD progression [26]. In addition, in male and female 3xTg-AD mice, chronic binge ethanol exposure for 5.5–9 months led to sustained downregulation of neuroprotective genes; additionally, this exposure increased the proinflammatory gene profile in the cortex of female mice, a pattern absent in males. Simultaneously, ethanol increased both total and hyperphosphorylated tau levels in the hippocampus and piriform cortex of females but not in males, suggesting elevated susceptibility to AD pathology in females [27]. Chronic ethanol exposure for ten weeks in APP/PS1 mice has been shown to result in increased brain atrophy, an augmented number of amyloid plaques, metabolic dysregulation, and dysregulated behaviors—an increase in locomotor activity and compulsive-like behavior [28]. Moreover, acute ethanol exposure increases amyloid-beta 40 (Aβ40) levels in the hippocampal interstitial fluid in these mice during withdrawal [29]. In addition to in vivo studies, several in vitro studies demonstrate that ethanol exposure upregulates BACE1 expression and subsequent Aβ production in human SK-N-MC neuroblastoma cells [28,30]. Additionally, in microglial cells, a 24 h exposure to alcohol diminishes Aβ1–42 phagocytosis, a mechanism potentially contributing to increased Aβ accumulation [28].

Recent studies have shifted focus beyond neurons as Aβ producers, highlighting astrocytes, which make up approximately 50–70% of the central nervous system (CNS) cell population, as potential sources of amyloid generation in AD [31,32]. We previously demonstrated the region-specific augmentation of astrocytic amyloids in rhesus macaques subjected to SIV infection or chronic morphine administration. In this study, we reported that the exposure of human primary astrocytes (HPAs) to morphine and HIV-1 Tat resulted in the upregulation of hypoxia-inducible factor (HIF-1α)-BACE1 and HIF-1α-lncRNA BACE1-AS axes, respectively, which, in turn, regulated the expression of the APP and Aβ. Further, these amyloid cargoes were disseminated extracellularly through astrocyte-derived extracellular vesicles both in vitro and in vivo, leading to neuronal injury [20,32,33,34]. However, until now, there have been no studies addressing the impact of alcohol on astrocytic amyloid generation and its potential subsequent contribution to neuroinflammation, which could result in cognitive deficits.

Collectively, our results demonstrate that ethanol exposure leads to lncRNA BACE1-AS-mediated astrocytic amyloid generation and dysregulated alcohol metabolism, leading to oxidative stress and neuroinflammation in HPAs. Thus, this study shows that lncRNA BACE1-AS could be harnessed as a therapeutic target for alcohol-mediated neurological deficit(s) mediated by amyloid accumulation.

## 2. Materials and Methods

### 2.1. Reagents

The following antibodies were used for this study: Aβ mOC64 (ab201060, Abcam, Waltham, MA, USA), APP (ab15272, Abcam, Waltham, MA, USA), BACE1 (ab263901, Abcam, Waltham, MA, USA), ADH (ab108197, Abcam, Waltham, MA, USA), ALDH2 (ab133306, Abcam, Waltham, MA, USA), CYP2E1 (ab28146, Abcam, Waltham, MA, USA), GFAP (G3893, Sigma-Aldrich, St. Louis, MO, USA), IL1β (ab9722, Abcam, Waltham, MA, USA), Anti-4 Hydroxynonenal antibody (ab46545, Abcam, Waltham, MA, USA), β-actin (A5316, Sigma-Aldrich, St. Louis, MO, USA), goat anti-mouse (sc-2005, Santa Cruz Biotechnology, Dallas, TX, USA), goat anti-rabbit (sc-2004, Santa Cruz Biotechnology, Dallas, TX, USA), Alexa Fluor 647-conjugated goat anti-mouse (A32728, ThermoFisher Scientific, Waltham, MA, USA), and Alexa Fluor 488-conjugated goat anti-mouse (A11001, ThermoFisher Scientific, Waltham, MA, USA).

### 2.2. Cell Culture

HPAs were acquired from the University of Washington, Seattle. Briefly, human fetal brain tissues were procured from voluntarily terminated pregnancies in partnership with the Birth Defects Research Laboratory at the University of Washington. The procedure adhered to both state and federal regulations. All individuals provided informed consent using an Institutional Review Board-approved consent form at the University of Washington. Cells were maintained in astrocyte medium (Catalog #1801, ScienCell Research Laboratories, Carlsbad, CA, USA) enriched with 2% fetal bovine serum (FBS, Catalog #0010, ScienCell Research Laboratories, Carlsbad, CA), astrocyte growth supplement (Catalog #1852, ScienCell Research Laboratories, Carlsbad, CA), and a penicillin–streptomycin mix (Catalog #0503, ScienCell Research Laboratories, Carlsbad, CA). For experimental assays, HPAs were plated at densities of 0.3 × 10^6^ cells/well in 6-well plates and 0.05 × 10^6^ cells/well in 24-well plates, ensuring that the number of passages did not exceed five. The cells were grown in a 5% CO_2_ humidified incubator at 37 °C overnight until they reached 70% confluency. Subsequently, HPAs were exposed to the specified dose of ethanol for the respective periods, as described in Section 3.

### 2.3. siRNA Transfection

HPAs were plated in 6-well plates at a density of 0.3 × 10^6^ cells/well and cultured overnight at 37 °C in a humidified 5% CO_2_ incubator. Upon reaching approximately 70% confluency, the cells were transfected with lncRNA BACE1-AS (customized by Integrated DNA Technologies, Coralville, IA, USA) or scrambled siRNA in Opti-MEM media (31985062, Life Technologies, Carlsbad, CA, USA) employing Lipofectamine RNAiMAX transfection reagent (13778150, ThermoFisher Scientific, Waltham, MA, USA) according to the manufacturer’s guidelines. After transfection, the cells were maintained for 12–18 h in a 5% CO_2_-containing humidified incubator at 37 °C. Post-transfection, the cells were exposed to 12.5 mM ethanol for 24 h. The efficacy of the gene knockdown was validated through qPCR.

### 2.4. Quantitative Real-Time PCR (qPCR)

Total RNA from HPAs was extracted utilizing the Quick-RNA MicroPrep kit (R1055, Zymo Research Corporation, Irvine, CA, USA,) according to the manufacturer’s instructions. The qPCR methodology was executed following established procedures [35]. Concisely, a total RNA amount equal to 1 ug was reverse-transcribed to cDNA with the Verso cDNA synthesis kit (AB1453/B, Thermo Fisher Scientific, Waltham, MA, USA), as per the manufacturer’s instructions. The qPCR assays utilized TaqMan probes, and specific primers for lncRNA BACE1-AS (Hs_04232267), BACE1 (Hs_01121195), APP (Hs_00169098), TNF-α (Hs_00174128), IL-1β (Hs_01555410), IL-6 (Hs_00 174131), and GAPDH (Hs_002786624) were acquired from Thermo Fisher Scientific, Waltham, MA, USA. Comparative expression levels were determined by normalizing mean CT values against the housekeeping gene GAPDH, and the 2^−ΔΔCT^ method was used to calculate gene expression.

### 2.5. Western Blot

The protein expression in the samples was assessed using a conventional Western blotting technique, as detailed in a prior study [36]. In brief, proteins extracted from HPAs were measured using the Pierce BCA Protein Assay Kit (product number 23227, Thermo Fisher Scientific, Waltham, MA, USA) according to the manufacturer’s instructions. SDS-PAGE was used for protein separation before transferring them to a PVDF membrane (product code IPVH00010, Millipore Sigma, St. Louis, MO, USA). For immuno-blotting, specific primary antibodies in combination with horseradish peroxidase-conjugated anti-mouse and anti-rabbit immunoglobulins were used. The resulting immunocomplexes were visualized using the SuperSignal chemiluminescent substrate (product code VJ311133, Thermo Fisher Scientific, Waltham, MA, USA) according to the manufacturer’s instructions. The protein levels were normalized to β-actin. The protein bands were digitally captured with a GT-X750 scanner (Seiko Epson Corporation, Nagano, Japan) and subsequently quantified using the Image J Launcher software (version 1.4.3.67, NIH, Bethesda, MD, USA).

### 2.6. Immunocytochemistry

HPAs were seeded onto 11 mm coverslips within a 24-well plate at a density of 0.05 × 10^6^ cells per well. The cells were cultured in a humidified 5% CO_2_ incubator at 37 °C for a duration of 24 h. After reaching confluency, the HPAs were treated with ethanol at a concentration of 12.5 mM for 24 h, followed by the fixation of the cells with 4% Paraformaldehyde (PFA) and incubation with GFAP (G3893, Sigma-Aldrich, St. Louis, MO, USA) and Aβ 647 (ab224025, Abcam, Waltham, MA, USA) antibodies, as described previously [32]. Fluorescence images were captured on a Zeiss Observer using a Z1 inverted microscope (Carl Zeiss, Thornwood, NY, USA) and analyzed using the AxioVs 40 Version 4.8.0.0 software (Carl Zeiss MicroImaging GmbH). The mean fluorescence intensity was quantified using the Image J Launcher software (version 1.4.3.67, NIH, Bethesda, MD, USA).

### 2.7. Statistical Analysis

The grouped data are represented as mean ± standard error of the mean (SEM). The statistical significance among the multiple experimental groups was assessed using one-way ANOVA followed by Tukey’s post hoc test, and Student’s *t*-test with the Mann–Whitney test was used to compare between two groups using the GraphPad Prism Software (Version 10). A *p*-value less than 0.05 was considered statistically significant.

## 3. Results

### 3.1. Alcohol Exposure Induces Astrocytic Amyloid Generation and Inflammation

We first aimed to evaluate the expression of toxic amyloid proteins and neuroinflammatory cytokines in HPAs. The HPAs were exposed to various ethanol concentrations (0, 6.25, 12.5, 25, 50, 100, 200 mM) for a period of 24 h. Subsequently, the cell lysates were analyzed to assess *APP*, *BACE1*, *lncRNA BACE1-AS*, and *interleukin-1β* (*IL1β*) expression. As depicted in Figure 1A–D, ethanol concentrations ranging from 6.25 to 200 mM notably increased (* *p* < 0.05) the mRNA levels of *APP*, *BACE1*, *lncRNA BACE1-AS*, and *IL1β* in a dose-dependent manner compared to the control cells. For further validation, we measured the protein levels of APP, Aβ mOC64 (detects monomeric, oligomeric, and fibrillar forms of Aβ1-42), BACE1, and the neuroinflammatory cytokine mature IL1β in HPAs exposed to the same ethanol concentrations. As shown in Figure 2A–D, significant (* *p* < 0.05) elevations in protein expression were observed in a dose-dependent manner when compared to control cells without ethanol exposure. Notably, the 12.5 mM ethanol concentration at 24 h was consistently associated with the significant upregulation of all markers and thus was chosen for further experimentation. This concentration of alcohol (12.5 mM) was associated with motor incoordination, slowed reaction times, and cognitive deficits in humans [37]. Interestingly, immunocytochemistry data demonstrated that there was a significant (* *p* < 0.05) increase in the expression of GFAP and Aβ mOC64 in HPAs exposed to a 12.5 mM dose of ethanol for 24 h (Figure 2E,F). Of note, Aβ1-42 is the primary toxic form of amyloid contributing to the neurodegeneration observed in Alzheimer’s disease [38].

Next, we sought to examine the time-dependent expression (at 3, 6, 12, 24, 48, 72, and 96 h post-ethanol exposure) of AD and inflammatory markers in astrocytes following exposure to 12.5 mM ethanol. As illustrated in Figure 3A–C, exposure to ethanol led to the significant upregulation (* *p* < 0.05) of the mRNA expression of *APP*, *BACE1*, and *lncRNA BACE1-AS* in HPAs, indicative of astrocytic amyloid generation in a time-dependent manner. Additionally, the proinflammatory cytokines *tumor necrosis factor-α (TNFα)*, *interleukin-6 (IL6)*, and *IL1β* were also increased (* *p* < 0.05) in a time-dependent manner after 12.5 mM ethanol exposure (Figure 3D–F). Moreover, ethanol at a concentration of 12.5 mM markedly elevated (* *p* < 0.05) the protein expression of the APP, Aβ mOC64, and BACE1 in a time-dependent manner in HPAs (Figure 3G–I).

Furthermore, the effects of ingested beverage alcohol, specifically ethanol, on various organs, notably the brain, depend on the concentration of ethanol reached in the bloodstream and the length of time over which it is consumed [39]. The primary metabolic pathway of ethanol involves its oxidation to acetaldehyde by alcohol dehydrogenases (ADHs), which is subsequently oxidized to acetate by mitochondrial aldehyde dehydrogenase 2 (ALDH2) [40]. Additionally, certain mitochondrial/microsomal enzymes, like cytochrome P450, notably CYP2E1, metabolize a minor portion of the consumed ethanol and play a pivotal role in reactive oxygen species (ROS) production [41,42,43]. An increase in ROS leads to oxidative stress and the generation of reactive and toxic aldehydes, including lipid peroxidation-derived α,β-unsaturated aldehydes such as 4-hydroxynonenal (4-HNE) and malondialdehyde [44]. To further explore the impact of ethanol exposure on the expression levels of alcohol-metabolizing enzymes, HPAs were exposed to an ethanol concentration of 12.5 mM. As illustrated in Figure 4A–C, there were significant increases (* *p* < 0.05) in the expression levels of ADH, ALDH2, and CYP2E1, showing a time-dependent pattern. Moreover, we also observed a significant elevation (* *p* < 0.05) in the level of 4-Hydroxynonenal (4-HNE) starting from six hours post-exposure (Figure 4D). Similarly, ethanol exposure significantly upregulated (* *p* < 0.05) the protein levels of the neuroinflammatory cytokine IL-1β (Figure 4E). Notably, a consistent increase was observed 24 h post-alcohol exposure in comparison to control cells. In light of these results, the optimal concentration and duration for alcohol’s effects were determined to be 12.5 µM for 24 h, which were subsequently selected for all further experiments.

### 3.2. Long Noncoding RNA BACE1-AS Regulates Ethanol-Mediated Amyloid Generation and Inflammation in Astrocytes

After establishing that ethanol can upregulate amyloid in astrocytes, we next wanted to assess the role of lncRNA BACE1-AS in amyloid generation, alcohol metabolism, and inflammation. HPAs were transfected with either scrambled or lncRNA BACE1-AS siRNA, followed by exposure to ethanol (12.5 mM, 24 h). As shown in Figure 5A–F, HPAs transfected with scrambled siRNA and subsequently exposed to ethanol exhibited a significant increase (* *p* < 0.05) in the mRNA expression of *lncRNA BACE1-AS*, *BACE1*, *APP*, *TNF-α*, *IL-6*, and *IL-1β*. There was also a notable increase in the protein levels of APP, AβmOC64, and BACE1 (* *p* < 0.05) compared to cells not exposed to ethanol (Figure 6A–C). Conversely, HPAs transfected with lncRNA BACE1-AS siRNA and then exposed to ethanol displayed a significant reduction (^#^
*p* < 0.05) in the mRNA expression levels of *lncRNA BACE1-AS*, *APP*, *BACE1*, *TNF-α*, *IL-6*, and *IL-1β* (Figure 5A–F). These reductions extended to the protein expression of APP, AβmOC64, and BACE1 when compared with HPAs transfected with scrambled siRNA and exposed to ethanol (Figure 6A–C).

Next, to elucidate the role of lncRNA BACE1-AS in the ethanol-mediated upregulation of alcohol-metabolizing enzymes and their contribution to oxidative stress and proinflammatory cytokine production, we assessed the protein expression levels of ADH, ALDH2, 4-HNE, and IL-1β. We demonstrated that HPAs transfected with scrambled siRNA, followed by ethanol exposure, exhibited a significant increase in the protein levels of ADH, 4-HNE, and IL-1β (* *p* < 0.05) compared to cells not exposed to ethanol (Figure 6D–G). Conversely, transfection with lncRNA BACE1-AS siRNA led to a significant reduction in the expression levels of ALDH2, ADH, 4-HNE, and mIL-1β in comparison to the ethanol-exposed, scrambled-siRNA-transfected group (^#^
*p* < 0.05) (Figure 6D–G).

### 3.3. Alcohol-Induced Astrocytic Amyloid Generation Leading to Oxidative Stress and Neuroinflammation Involves lncRNA BACE1-AS

Exposure to alcohol leads to the increased expression and accumulation of the neurotoxic Aβ precursor protein—APP—and the cleaving enzyme BACE1 in HPAs. In parallel to the production of amyloids, key alcohol-metabolizing enzymes—ADH, ALDH2, and CYP2E1—were also shown to be upregulated by alcohol exposure in astrocytes. Further, in the present study, it was demonstrated that lncRNA BACE1-AS can regulate the APP and BACE1 along with alcohol-metabolizing enzymes in astrocytes. After translation, both proteins are present in the endoplasmic reticulum, trans-Golgi network, endosome, and plasma membrane, where they contribute to Aβ production [6,45]. Following its generation, Aβ accumulates in different cellular organelles, such as mitochondria [46], the ER [47,48,49], the trans-Golgi network (TGN) [50], post-TGN secretory vesicles [47], endosomes [51], lysosomes [52], multivesicular bodies (MVBs) [53], and the cytosol [54,55]. Moreover, these neurotoxic amyloids in the mitochondria and endoplasmic reticulum, in coordination with alcohol metabolites like acetaldehyde, lead to oxidative stress and inflammation [56,57,58]. Additionally, in turn, via a positive feedback mechanism, ROS may activate lncRNA BACE1-AS and heighten the response to alcohol, thus implicating the therapeutic potential of lncRNA BACE1-AS inhibitors (Figure 7).

## 4. Discussion

Heavy alcohol consumption constitutes a significant etiological factor in morbidity and mortality profiles worldwide [59]. The impact of ≤2.2-drink alcohol (moderate level) intake (per day) on cognitive functions and the risk of developing dementia are subjects of controversial discussion, particularly regarding the impact of low-dose alcohol [60,61,62]. An earlier investigation has underscored the role of astrocytes in mediating inflammation and oxidative stress associated with alcohol consumption [63]. Additionally, research employing the single-cell technique has further elucidated the involvement of astrocytes in neuroinflammation in the human alcoholic brain, underscoring their significant impact on brain health and disease states [64]. The present study underscores the novel role of astrocytes in the progression of alcohol-induced amyloid generation, leading to neuroinflammation. Our findings reveal that ethanol exposure to HPAs leads to the upregulation of the neurotoxic Aβ1-42. Additionally, alcohol modulates the expression of other AD markers—APP and BACE1; an oxidative stress marker—4-HNE; alcohol-metabolizing enzymes—ADH, ALDH2, and CYP2E1; and proinflammatory cytokines (*TNF-α*, *IL1β*, *IL6*). Furthermore, our research has demonstrated that lncRNA BACE1-AS regulates the expression of AD markers, proinflammatory cytokines, and enzymes involved in alcohol metabolism. These findings contribute to the growing body of evidence on the complex interactions between alcohol consumption and the development of Alzheimer’s-like pathology, emphasizing the critical role of astrocytes in this context. The involvement of astrocytes in amyloid generation and neuroinflammation has been reported by us [32] and others [65]. Reactive astrocytes, which can be induced by a myriad of factors, including inflammatory cytokines and substance use, have been shown to contribute to the production of β-amyloid [66,67]. Specifically, in the APPswe/PS1dE9 double-transgenic mouse model, astrocytes exhibit the upregulation of genes associated with inflammatory responses [68]. These studies underscore the critical role of astrocytes in the propagation of neuroinflammatory pathways in AD. Our current research extends these findings by exploring the impact of ethanol exposure on HPAs. We demonstrated that ethanol exposure results in the upregulation of toxic amyloid forms, highlighting the influence of alcohol on astrocytic amyloidosis (Figure 2B,E). Furthermore, our investigation revealed the presence of Aβ1-42 signals in the nucleus in addition to the cytoplasm. One plausible explanation is that the translocation of Aβ1-42 into the nucleus may signify a transcriptional response to alcohol exposure. Moreover, earlier studies have provided evidence supporting the notion that Aβ1-42 acts as a transcriptional regulator [69,70]. Therefore, the localization of Aβ1-42 within the nucleus warrants further exploration to elucidate its potential role in transcriptional regulation, particularly in the context of alcohol-induced responses. Additionally, our results indicate that alcohol not only increases the expression of APP and BACE1 translationally but also induces the transcriptional activation of these genes (Figure 1A,B and Figure 2A,C). These results are in alignment with previous research that highlights the crucial role of astrocytes in the progression of AD [31,71,72,73]. Our findings also demonstrate that alcohol elevates the levels of proinflammatory cytokines (*TNF-α*, *IL-1β*, *IL-6*) at the transcriptional level (Figure 3D–F). Additionally, we observed an increase in the oxidative stress marker 4-HNE and the inflammatory cytokine mIL-1β at the translational level (Figure 4D,E). All of these changes may contribute to the pathogenesis of neuroinflammation, a critical factor in the development of neurodegenerative diseases. Our results are in agreement with previous research demonstrating alcohol-mediated neuroinflammation and oxidative stress [74,75], further substantiating alcohol-induced alterations in the brain. Nevertheless, the role of alcohol in relation to Alzheimer’s-like pathology, specifically in regard to astrocytes, has not been documented earlier. Our study thus provides novel insights into how alcohol may contribute to astrocyte-mediated amyloidogenesis and neuroinflammation, underscoring the complex interactions between alcohol and genetic factors such as AD- and alcohol-metabolism-associated genes in the development and progression of AD. Of note, ethanol exposure has been shown to suppress the phagocytic capability of microglia for Aβ1-42, a critical process in the clearance and management of amyloid plaques in the brain. Additionally, ethanol exposure increases the levels of several complement transcripts in microglia [28]. The upregulation of these complement components could contribute to alcohol-induced neuropathology by enhancing inflammation, potentially facilitating the development and progression of AD [26,76]. While our investigation primarily focuses on astrocytic amyloidosis, the critical role of enzymes involved in alcohol metabolism cannot be disregarded. Ethanol is metabolized and eliminated via multiple pathways, with ALDH, ADH, CYP2E1, and catalase playing pivotal roles in this process [77]. These enzymes convert ethanol into less toxic compounds, facilitating their clearance from the body [78]. Importantly, our study found the significant upregulation of ADH, ALDH2, and CYP2E1 within astrocytes in response to ethanol exposure, indicating their critical function in the metabolic processing of alcohol within these glial cells (Figure 4A–C). Furthermore, Jin et al. (2021) demonstrated the role of astrocytic ALDH2 in contributing to motor impairments associated with ethanol intoxication [79]. Moreover, previous studies have documented the activity of ADH and CYP2E1 in metabolizing ethanol within specific neuronal populations throughout various brain regions [80]. Prior studies and ours underscore the notion that alcohol metabolism extends beyond the liver, occurring within the brain and impacting astrocyte functionality directly. A groundbreaking aspect of our research is the identification of ADH within astrocytes. This discovery positions our study at the forefront of exploring alcohol metabolism in the brain, particularly in non-neuronal cells. Moreover, the involvement of BACE1 in the pathogenesis of amyloidosis associated with AD has been well established in the literature [81,82]. Expanding upon these findings, our prior investigations have elucidated the complex interplay between lncRNA BACE1-AS and BACE1 in the pathway leading to astrocytic amyloidosis, particularly in the context of HIV-Tat exposure [32]. The current study demonstrates a significant upregulation of lncRNA BACE1-AS following alcohol exposure (Figure 1C and Figure 3C). Employing gene-silencing techniques, we have validated the regulatory role of lncRNA BACE1-AS in AD markers, proinflammatory cytokines, and oxidative stress markers. Notably, we observed a marked reduction in APP, BACE1, and Aβ levels, as well as decreased levels of proinflammatory cytokines and the oxidative stress marker, in HPAs following lncRNA BACE1-AS silencing and/or ethanol treatment in comparison to ethanol-only treated control cells. This suggests a protective role of lncRNA BACE1-AS in alcohol-induced amyloid generation and neuroinflammatory and oxidative stress responses. Intriguingly, the expression of lncRNA BACE1-AS has been observed to increase in various types of cancer and cardiovascular disease [83,84,85,86], underscoring its potential role beyond neurodegenerative disorders. Additionally, recent studies have implicated lncRNA BACE1-AS as a contributor in Parkinson’s disease [87,88], and this lncRNA has been proposed as a blood-based biomarker for Alzheimer’s disease [89]. The convergence of these findings with our own underscores the multifaceted role of lncRNA BACE1-AS across a spectrum of diseases, suggesting its significance not only in the context of AD and astrocytic amyloidosis but also in a broader pathological landscape. Our study pioneers the exploration of lncRNA BACE1-AS in regulating the expression of alcohol-metabolizing enzymes, including ADH and ALDH2. We noted a substantial decrease in the expression of these enzymes in cells subjected to lncRNA BACE1-AS silencing, with or without ethanol exposure, suggesting a novel regulatory mechanism of alcohol metabolism mediated by lncRNA BACE1-AS (Figure 6D,E). Therefore, our findings not only reinforce the pivotal role of lncRNA BACE1-AS in the pathogenesis of astrocytic amyloidosis but also unveil its novel function in regulating the expression of key alcohol-metabolizing enzymes. Our comprehensive investigations reveal that alcohol exposure induces the expression of *lncRNA BACE1-AS*, *BACE1*, *APP*, Aβ moC64, *TNFα*, *IL1β*, and *IL6* as well as alcohol-metabolizing enzymes such as ADH, ALDH2, and CYP2E1 in astrocytes. The mechanism involves the upregulation of lncRNA BACE1-AS, which, in turn, contributes to neurotoxic Aβ1-42 formation, leading to astrocyte activation and the subsequent production of proinflammatory cytokines. Crucially, our studies elucidate the regulatory role of lncRNA BACE1-AS in the expression of alcohol-metabolizing enzymes in response to ethanol exposure. Previously, no scientific studies had established a relationship between alcohol-associated astrocytic amyloidosis and the involvement of lncRNA BACE1-AS in this process. This study, for the first time, demonstrates the role of lncRNA BACE1-AS in alcohol-induced astrocytic amyloidosis and the metabolism of alcohol. This study adds to our growing body of knowledge on the intricate relationship between alcohol consumption, neuroinflammation, and neurodegeneration, proposing that metabolic processes within astrocytes might provide novel insights into the pathogenesis of alcohol-related neurological disorders.

Furthermore, lncRNAs (longer than 200 nucleotides) are non-protein-coding transcripts that have been identified as important contributors to various cellular processes, such as the regulation of chromosome organization, replication, transcription, pre-mRNA splicing, and translation [90]. Likewise, lncRNA BACE1-AS has gained attention due to its regulatory role in a plethora of cellular processes [91]. One of the important roles of lncRNA BACE1-AS is its ability to increase the stability of the BACE1 transcript through the formation of RNA duplexes [92]. Additionally, in SH-SY5Y cells and AD transgenic mice, lncRNA-BACE1-AS has been demonstrated to modulate autophagy through the miR-214-3p/ATG5 signaling axis [93]. Our results demonstrate that lncRNA BACE1-AS regulates ethanol-metabolizing enzymes and AD markers, probably by transcriptional stability or post-transcriptional mechanisms. Nevertheless, future studies are required to establish the mechanistic links between lncRNA BACE1-AS, AD markers, and ethanol-triggered pathways.

Moreover, it is important to note that our research is primarily based on in vitro studies, which presents a limitation regarding the direct applicability of these findings to the in vivo context. Future studies, particularly those employing in vivo models, are essential to validate these in vitro observations and to fully understand their implications for the development and progression of alcohol-induced cognitive and neurological deficits.

## Figures and Tables

**Figure 1 cells-13-01173-f001:**
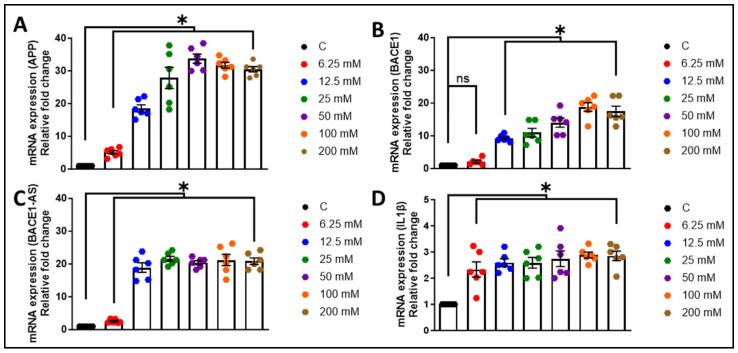
Ethanol induces the generation of amyloids and proinflammatory cytokines. qPCR analysis demonstrating mRNA levels of *APP* (**A**), *BACE1* (**B**), and *lncRNA BACE1-AS* (**C**) and *proinflammatory cytokine IL1β* (**D**) in HPAs following exposure to ethanol. GAPDH was utilized as an internal control. The data are presented as mean ± standard error of the mean (SEM); *n* = six biological replicates per group. One-way ANOVA with Tukey’s post hoc test was applied to assess statistical significance between multiple groups: * *p* < 0.05 vs. control. Abbreviations: APP: amyloid precursor protein; BACE1: β-site-cleaving enzyme; BACE1-AS: BACE1-antisense transcript; IL1β, interleukin-1β; GAPDH: glyceraldehyde 3-phosphate dehydrogenase; HPA: human primary astrocyte; qPCR: quantitative polymerase chain reaction.

**Figure 2 cells-13-01173-f002:**
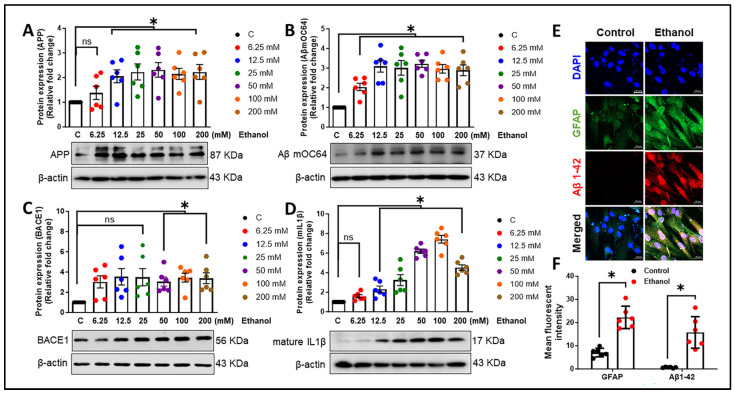
Ethanol induces the generation of amyloids and proinflammatory cytokines. Western blot analysis demonstrating protein levels of the APP (**A**), Aβ mOC64 (**B**), and BACE1 (**C**)—as well as the proinflammatory cytokine IL1β (**D**)—in HPAs following exposure to ethanol. β-Actin was utilized as an internal control. (**E**) Representative fluorescent photomicrographs showing increased expression of GFAP and Aβ1–42 proteins in HPAs in the presence or absence of ethanol (12.5 mM; 24 h). Scale bar, 20 μm. (**F**) The mean fluorescence intensity of GFAP and Aβ1–42. The data are presented as mean ± SEM; *n* = six biological replicates per group. One-way ANOVA with Tukey’s post hoc test and unpaired Student’s *t*-test with the Mann–Whitney test were applied to assess statistical significance: * *p* < 0.05 versus control. Abbreviations: APP, amyloid precursor protein; Aβ, amyloid beta; BACE1, β-site-cleaving enzyme; mIL1β, mature interleukin-1β; GAPDH, glyceraldehyde 3-phosphate dehydrogenase; HPA, human primary astrocyte; qPCR, quantitative polymerase chain reaction.

**Figure 3 cells-13-01173-f003:**
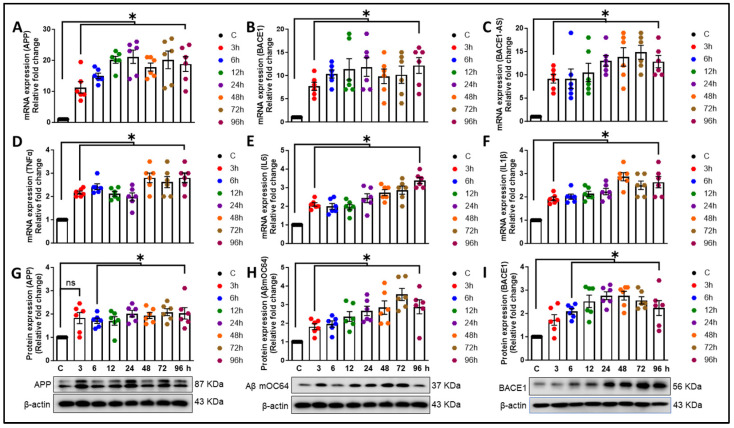
Ethanol mediated the expression of AD markers and proinflammatory cytokines in a time-dependent manner in HPAs. qPCR (**A**–**F**) analysis showing the time-dependent (3–96 h) mRNA expression of *APP* (**A**), *BACE1* (**B**), *BACE1-AS* (**C**), and the proinflammatory cytokines *TNFα* (**D**), *IL6* (**E**), and *IL1β* (**F**) in HPAs exposed to ethanol at a 12.5 mM dose for the indicated times. GAPDH was used as an internal control. Western blots (**G**–**I**) showing the time-dependent upregulation of APP (**G**), Aβ mOC64 (**H**), and BACE1 (**I**) proteins in HPAs following the exposure of cells to ethanol (12.5 mM) for the indicated times. For the APP and Aβ mOC64, a single membrane was divided into two halves: the upper half was probed with the APP antibody, and the lower half was probed with the Aβ mOC64 antibody; β-actin was similar for both of these blots. β-Actin was used as an internal control for normalization. Data are presented as mean ± SEM; *n* = six biological replicates per group. One-way ANOVA followed by Tukey’s post hoc test was used to determine the statistical significance: * *p* < 0.05 versus control. Abbreviations: APP, amyloid precursor protein; Aβ, amyloid beta; BACE1, β-site-cleaving enzyme; GAPDH, glyceraldehyde 3-phosphate dehydrogenase; BACE1-AS, BACE1-antisense transcript; TNFα, tumor necrosis factor α; IL6, interleukin-6; IL1β, interleukin-1β; HPA, human primary astrocyte; qPCR, quantitative polymerase chain reaction.

**Figure 4 cells-13-01173-f004:**
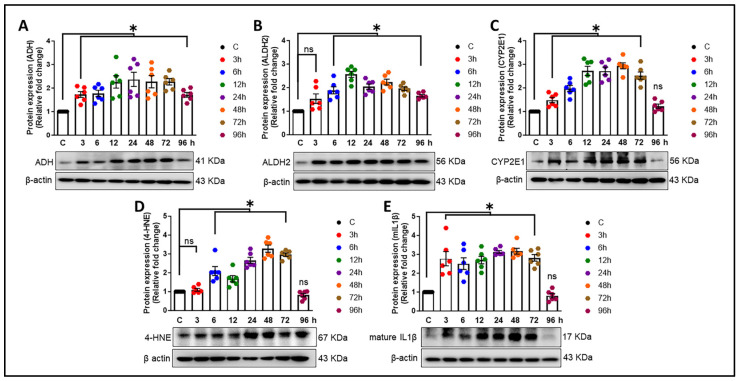
Ethanol mediated the expression of alcohol-metabolizing enzymes, an oxidative stress marker, and proinflammatory cytokines in HPAs. Western blot analysis showing the time-dependent (3–96 h) upregulation of ADH (**A**), ALDH2 (**B**), CYP2E1 (**C**), and the oxidative stress marker 4-HNE (**D**), as well as the proinflammatory cytokine IL1β (**E**), in HPAs exposed to ethanol at a 12.5mM dose for the indicated times. β-Actin was used as an internal control. Data are presented as mean ± SEM; *n* = six biological replicates per group. One-way ANOVA followed by Tukey’s post hoc test was used to determine the statistical significance: * *p* < 0.05 versus control. Abbreviations: ADH, alcohol dehydrogenase; ALDH2, aldehyde dehydrogenase 2; CYP2E1, cytochrome P450 2E1; 4-HNE, 4-hydroxynonenal; mIL1β, mature interleukin 1β; HPA, human primary astrocyte.

**Figure 5 cells-13-01173-f005:**
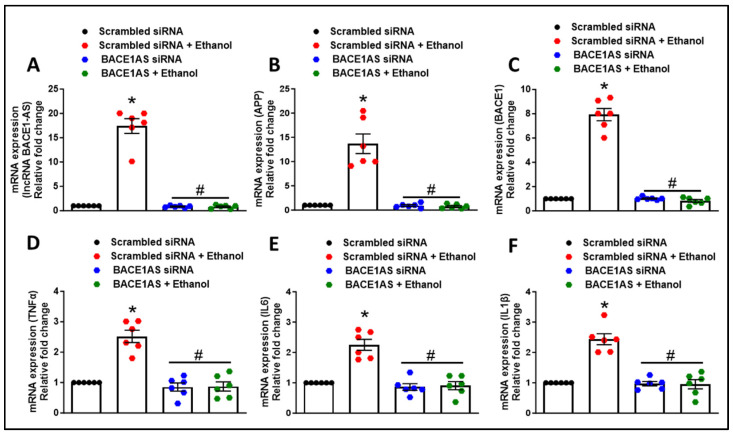
Role of lncRNA BACE1-AS in ethanol-induced astrocytic amyloidosis and proinflammatory cytokine production. qPCR showing mRNA expression of *lncRNA BACE1-AS* (**A**), *APP* (**B**), and *BACE1* (**C**), as well as the proinflammatory cytokines *TNFα* (**D**), *IL6* (**E**), *IL1β* (**F**), in HPAs transfected with either BACE1-AS or scrambled siRNA following exposure to ethanol (12.5 mM; 24 h). GAPDH was utilized as an internal control. The data are presented as mean ± SEM; *n* = six biological replicates per group. One-way ANOVA with Tukey’s post hoc test was applied to assess statistical significance: * *p* < 0.05 versus control, *^#^ p* < 0.05 versus ethanol. Abbreviations: APP, amyloid precursor protein; Aβ, amyloid beta; BACE1, β-site-cleaving enzyme; BACE1-AS, BACE1-antisense transcript; TNFα, tumor necrosis factor α; IL-6, interleukin-6; IL1β, interleukin-1β; HPA, human primary astrocyte; qPCR, quantitative polymerase chain reaction.

**Figure 6 cells-13-01173-f006:**
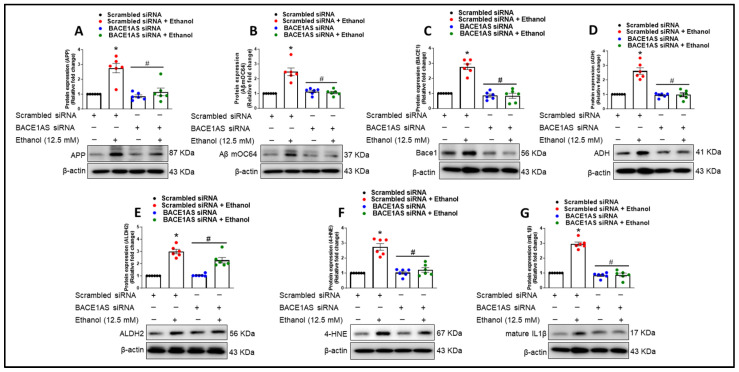
The role of lncRNA BACE1-AS in ethanol-induced astrocytic amyloidosis, alcohol metabolism, oxidative stress, and proinflammatory cytokine production. Western blot analysis demonstrating the expression of APP (**A**), Aβ mOC64 (**B**), and BACE1 (**C**), ADH (**D**), ALDH2 (**E**), and the oxidative marker 4-HNE (**F**), as well as the proinflammatory cytokine IL1β (**G**), in HPAs transfected with either lncRNA BACE1-AS or scrambled siRNA. β-Actin was utilized as an internal control. For BACE1 (**C**) and mIL1β (**G**), after developing mILb, the same membrane was reprobed with BACE1; thus, both proteins have similar β-actin. The data are presented as mean ± SEM; *n* = six biological replicates per group. One-way ANOVA with Tukey’s post hoc test was applied to assess statistical significance: ** p* < 0.05 versus control, ^#^
*p* < 0.05 versus ethanol. Abbreviations: APP, amyloid precursor protein; Aβ, amyloid beta; BACE1, β-site-cleaving enzyme; BACE1-AS, BACE1-antisense transcript; ADH, alcohol dehydrogenase; ALDH2, aldehyde dehydrogenase 2; 4-HNE, 4-hydroxynonenal; mIL1β, mature interleukin-1β; HPA, human primary astrocyte.

**Figure 7 cells-13-01173-f007:**
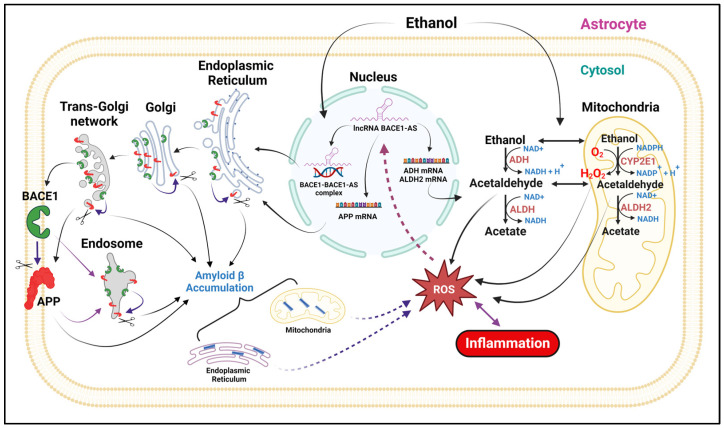
Schematic representation of alcohol-induced astrocytic amyloid generation leading to oxidative stress and neuroinflammation, mediated by lncRNA BACE1-AS. Created with Biorendor.com (https://app.biorender.com/user/signin, accessed on 4 July 2024). Abbreviations: APP, amyloid precursor protein; Aβ, amyloid beta; BACE1, β-site-cleaving enzyme; long noncoding RNA (lncRNA) BACE1-AS, BACE1-antisense transcript; ADH, alcohol dehydrogenase; ALDH2, aldehyde dehydrogenase 2; CYP2E1, cytochrome P450 2E1; ROS, reactive oxygen species; NAD^+^, nicotinamide adenine dinucleotide; NADP^+^, nicotinamide adenine dinucleotide phosphate; H_2_O_2_, hydrogen peroxide.

## Data Availability

Data will be available upon reasonable request from the corresponding author.

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
