# Peer review of "Astrocytes in Amyloid Generation and Alcohol Metabolism: Implications of Alcohol Use in Neurological Disorder(s)"

_cells, 2024, doi:10.3390/cells13141173_

Round 1
Reviewer 1 Report
Comments and Suggestions for Authors
In this manuscript, Kumar et al explore the role of astrocytes in alcohol-associated amyloid pathology using human primary astrocytes (HPA). They show that treatment of HPA with ethanol induced expression of lncRNA BACE1 antisense transcript (AS), neuroinflammation, amyloid generation and oxidative stress. Whereas the experimental data look robust, a mechanistic explanation of the proposed model, summarized in Figure 7, has a few conceptual gaps. For example, why/how lncRNA BACE1-AS affects both ALDH- and APP mRNA, is there any sequence similarity between them? Are there other predicted targets of this lncRNA-AS worth checking? Why are APP and BACE1 proteins not shown as transmembrane? Is gamma-secretase involved in the process? Where is Abeta getting accumulated leading to ROS formation? The discussion could benefit from a deeper analysis of the mechanistic links between lncRNA BACE1-AS and ethanol-triggered pathways.
Another question is, what kind of Abeta is detected by Abeta mOC64 antibody? In Fig.2 B it detects a 37kDa band or a duplet on Western blot, what form of Abeta is this? Is this Abcam antibody suitable for Western blot? the manufacturer only lists IHC-P, Dot and IHC-FrFl as suitable applications. In the discussion (line 384) ethanol-upregulated Abeta is referred to as “neurotoxic Abeta42”. Is Abeta 40 also affected? Is this Abeta accumulating intracellularly or it can be detected in conditioned HPA cell culture media?
To what extent the ethanol-induced oxidative stress is fuelled by Abeta? If BACE1 or gamma-secretase is pharmacologically blocked, will ethanol-induced oxidative stress still be detectable in the absence of Abeta?
Lines 401-403: it’s not quite clear what the Abeta42 signals in the nucleus are? is there any experimental evidence of Abeta42 translocation to the nucleus?
The beginning of Discussion from line 362 to line 381 mentioning epidemiological studies and mouse work is not directly relevant to the results of the current study and belongs to the Introduction.
It is unclear if there was variation between the batches of HPA coming from different donors. How was the inter donor variation addressed in the experimental design?
Small remarks:
Line 47: what concentration of alcohol is referred to in this study (ref [7])?
Line 56: “sex-specific” manner is probably more accurate
Line 100: what molecule was upregulated? Seemingly a coma is missing between HIF-1a and lncRNA BACE1-AS.
Line 215: what is meant by a “group”, technical replicates or individual cultures? Some bars seem to contain more than 6 datapoints e.g. Fig.1D 100 and 200mM ethanol.
Line 342: check spelling of “Alcohol-induced”.
Author Response
We are thankful to the editor and reviewer for the constructive comments. For better visibility of the individual replicates, we have edited all the figures. We have addressed each of the comments and highlighted the changes in yellow in the main text.
Reviewer 1:
In this manuscript, Kumar et al explore the role of astrocytes in alcohol-associated amyloid pathology using human primary astrocytes (HPA). They show that treatment of HPA with ethanol induced expression of lncRNA BACE1 antisense transcript (AS), neuroinflammation, amyloid generation and oxidative stress. Whereas the experimental data look robust, a mechanistic explanation of the proposed model, summarized in Figure 7, has a few conceptual gaps.
- For example, why/how lncRNA BACE1-AS affects both ALDH- and APP mRNA, is there any sequence similarity between them?
Response: We are thankful to the reviewer for the interesting question. It is well known in the literature that BACE1-AS can regulate the BACE1 expression, which further can cleave APP [1,2]. However, to the best of our knowledge, there is no evidence of direct interaction of lncRNA BACE1-AS with either ALDH or APP mRNA, nor any sequence similarity between them. However, our study demonstrates that lncRNA BACE1-AS regulates the expression of both APP and ALDH at the protein level. The mechanistic role of BACE1-AS in regulating different genes is part of a separate project & will be addressed in future studies.
- Are there other predicted targets of this lncRNA-AS worth checking?
Response: The reviewer has raised an exciting point. We would like to mention that lncRNA BACE1-AS has been shown to regulate different targets, including miRNAs and autophagy proteins [2,3], however, the mechanistic role of BACE1-AS in regulating different genes is a part of a separate project, & will be addressed in future studies.
- Why are APP and BACE1 proteins not shown as transmembrane?
Response: Although APP and BACE1 are type I transmembrane proteins, only a minor fraction of them can be observed on the plasma membrane. The majority of APP is localized in early endosomes, while BACE1 is predominantly situated in the late Golgi apparatus. Additionally, both proteins are present, in the endoplasmic reticulum, trans-Golgi network, endosome, and plasma membrane [4,5]. We have added a new schematic (Figure. 7) incorporating the different sites of localization for both APP and BACE1.
- Is gamma-secretase involved in the process?
Response: It is well known that gamma-secretase plays a critical role in Aβ42 production [5]. We will consider addressing the role of γ-secretase in future studies.
- Where is Abeta getting accumulated leading to ROS formation?
Response: Aβ has been reported to be accumulated in the different cellular organelles such as mitochondria [6], ER [7-9], Trans-Golgi network (TGN) [10] and post-TGN secretory vesicles [7], endosomes [11], lysosomes [12], multivesicular bodies (MVB) [13], and cytosol [14,15]. Furthermore, ROS also has been demonstrated to be produced in the different cellular compartments, including mitochondria, Endoplasmic reticulum, and cytoplasm [16]. Mitochondria is a prime source of endogenous ROS due to its main role in oxidative ATP production [16]. Furthermore, earlier studies have demonstrated the direct associations between ROS production and amyloid plaques in both transgenic mice and in human brain tissue in AD [17][18]. This study does not focus on the sub-cellular localization of Aβ & ROS. (Page no. 11-12; Line no. 378-392)
- The discussion could benefit from a deeper analysis of the mechanistic links between lncRNA BACE1-AS and ethanol-triggered pathways.
Response: We are thankful to the reviewer for very important question. Long-non-coding RNAs- lncRNAs (longer than 200 nucleotides) are non-protein coding transcripts that have been identified as important contributors to various cellular processes, such as the regulation of chromosome organization, replication, transcription, pre-mRNA splicing, and translation [19]. Likewise, the lncRNA BACE1-AS has gained attention due to its regulatory role in a plethora of cellular processes [1]. One of the important roles of lncRNA BACE1-AS is its ability to increase the stability of the BACE1 transcript through the formation of RNA duplexes [20]. Additionally, In SH-SY5Y cells and AD transgenic mice, lncRNA-BACE1-AS has been demonstrated to modulate autophagy through the miR-214-3p/ATG5 signaling axis [3]. Our results demonstrate that lncRNA BACE1-AS regulates the ethanol metabolizing enzymes and AD markers, by probable transcriptional stability or post-transcriptional mechanisms. However, the mechanistic links between lncRNA BACE1-AS, AD markers, and ethanol-triggered pathways, are part of a separate study and will be addressed in the future (Page no. 15; Line no. 518-530)
- Another question is, what kind of Abeta is detected by Abeta mOC64 antibody? In Fig. 2B it detects a 37kDa band or a duplet on Western blot, what form of Abeta is this? Is this Abcam antibody suitable for Western blot? the manufacturer only lists IHC-P, Dot and IHC-FrFl as suitable applications.
Response: The Abeta mOC64 antibody detects Abeta1-42. In Western blot [21,22] and dot blot analyses, this antibody recognizes monomeric, oligomeric and fibrillar forms of beta-Amyloid 1-42 of various molecular weights (https://www.abcam.com/products/primary-antibodies/beta-amyloid-1-42-antibody-moc64-ab201060.html#lb). Furthermore, multiple post-translational modifications of Aβ have been identified. Additionally, Aβ1‐42 forms different oligomeric Aβ species, ranging from low (2–4mers) to high-n oligomers (12–48mers), which vary in their molecular weight [23]. Therefore, this antibody can bind to different forms of Aβ with varied molecular weights. However, for our study, we focused on the 37-40 kDa band based on previous publications [24,25]. We acknowledge that the reviewer is correct that the manufacturer only lists IHC-P, Dot, and IHC-FrFl as suitable applications for this antibody. Additionally, we would like to mention that there are numbers of reports available where this antibody has been used for Western blots and authors have focused on the 37-40 kDa bands of Aβ [21,22,26-29]. (Page no. 6; Line no .221)
Dot blot analysis of human beta Amyloid 1-42 labeled with ab201060 at 1/7000 dilution.
Lane 1: beta Amyloid (Aβ) 1-40.
Lane 2: beta Amyloid (Aβ) 1-42.
(Abcam website: https://www.abcam.com/products/primary- antibodies/beta-amyloid-1-42-antibody-moc64-ab201060.html#lb)
- In the discussion (line 384) ethanol upregulated Abeta is referred to as “neurotoxic Abeta42”.
Response: In this manuscript, we have used the Abeta mOC64 antibody, which detects the Abeta1-42 form (https://www.abcam.com/products/primary-antibodies/beta-amyloid-1-42-antibody-moc64-ab201060.html#lb), the sole focus of this study. It is well established that Abeta1-42 is the primary toxic form of amyloid contributing to the neurodegeneration observed in Alzheimer's Disease [30]. Therefore, we mentioned Abeta42 as neurotoxic. (Page no. 15; Line no. 507)
- Is Abeta 40 also affected? Is this Abeta accumulating intracellularly, or it can be detected in conditioned HPA cell culture media?
Response: We are thankful to the reviewers for the interesting question. There are two major isoforms of amyloid-beta (Aβ): the 42-residue Aβ42 and the 40-residue Aβ40. Furthermore, Aβ42 is the neurotoxic form and the major component of amyloid plaques, with Aβ40 present to a lesser extent. Additionally, Aβ42 has been shown to accumulate in astrocytes [31,32]. Nevertheless, we cannot deny the possibility of Aβ40 in intracellular accumulation or in the human astrocyte-conditioned media. However, in this study, we solely focused on Aβ42, and the role of Aβ40 will be addressed in future studies.
- To what extent the ethanol-induced oxidative stress is fueled by Abeta? If BACE1 or gamma-secretase is pharmacologically blocked, will ethanol-induced oxidative stress still be detectable in the absence of Abeta?
Response: A very important point has been raised by the reviewer. Earlier evidence supports that in the neuroblastoma cell line SK-N-MC, ethanol-induced reactive oxygen species (ROS) were responsible for Aβ production via the induction of BACE1 [33]. Additionally, it is well known that ethanol can directly cause mitochondrial dysfunction and oxidative stress in non-neural cells [34-36]. Therefore, it is possible that ethanol can directly induce oxidative stress in the absence of Aβ. However, the extent to which ethanol-induced oxidative stress is mediated by Aβ warrants further investigation. Additionally, the role of BACE1 or gamma-secretase in Aβ-mediated ethanol-induced oxidative stress will be addressed in future studies.
- Lines 401-403: it’s not quite clear what the Abeta42 signals in the nucleus are? is there any experimental evidence of Abeta42 translocation to the nucleus?
Response: Our plausible explanation is that the translocation of Aβ1-42 into the nucleus may signify a transcriptional response to alcohol exposure via lncRNA BACE1-AS. Moreover, accumulated studies have provided evidence supporting the notion that Aβ1-42 can be translocated in the nucleus [37,38] & functions as a transcriptional regulator (Ref [68-69] in the original manuscript).
- The beginning of Discussion from line 362 to line 381 mentioning epidemiological studies and mouse work is not directly relevant to the results of the current study and belongs to the Introduction.
Response: We are thankful for the reviewer's suggestion. Appropriate changes have been made in the manuscript. (Page no. 2-3; Line no. 44-46, 77-82, 97-101)
- It is unclear if there was variation between the batches of HPA coming from different donors. How was the inter donor variation addressed in the experimental design?
Response: We are thankful for the reviewer's question. We would like to mention that each batch of HPAs was obtained from a different random donor. After receiving the cells, we performed staining with astrocyte-specific marker- GFAP to ensure purity, and only cells that were found to be >98% pure were used for subsequent experiments. Furthermore, to address inter-donor variation, we performed multiple replicates of each experiment using samples from each donor. Additionally, we standardized our protocols to minimize technical variability. All samples were processed under the same conditions, ensuring consistency in sample handling.
Small remarks:
- Line 47: what concentration of alcohol is referred to in this study (ref [7])?
Response: ≥450 g of alcohol per week (Page no 2., Line no 48.)
- Line 56: “sex-specific” manner is probably more accurate.
Response: Done (Page no 2., Line no. 58)
- Line 100: what molecule was upregulated? Seemingly a coma is missing between HIF-1a and lncRNA BACE1-AS.
Response: Done (Page no 3., Line no 113.)
- Line 215: what is meant by a “group”, technical replicates, or individual cultures? Some bars seem to contain more than 6 datapoints e.g. Fig.1D 100 and 200mM ethanol.
Response: A group here represents six biological replicates for each concentration of ethanol. Furthermore, we would like to mention that each bar has six data points, and Figure 1D- 100 and 200mM ethanol, as pointed out by the reviewer, also contains six data points. Indeed, it appears that there are more than six data points in Figure 1D for the 100 mM and 200 mM ethanol concentrations. This may be due to the superimposition of data points with the error bar. For better understanding, required changes have been made in the data. Enlarged figures are enclosed for confirmation.
- Line 342: check spelling of “Alcohol-induced”.
Response: Done (Page no 12., Line no. 373)
References
- Liu, T.; Huang, Y.; Chen, J.; Chi, H.; Yu, Z.; Wang, J.; Chen, C. Attenuated ability of BACE1 to cleave the amyloid precursor protein via silencing long noncoding RNA BACE1‑AS expression. Mol Med Rep 2014, 10, 1275-1281, doi:10.3892/mmr.2014.2351.
- Sayad, A.; Najafi, S.; Hussen, B.M.; Abdullah, S.T.; Movahedpour, A.; Taheri, M.; Hajiesmaeili, M. The Emerging Roles of the beta-Secretase BACE1 and the Long Non-coding RNA BACE1-AS in Human Diseases: A Focus on Neurodegenerative Diseases and Cancer. Front Aging Neurosci 2022, 14, 853180, doi:10.3389/fnagi.2022.853180.
- Zhou, Y.; Ge, Y.; Liu, Q.; Li, Y.X.; Chao, X.; Guan, J.J.; Diwu, Y.C.; Zhang, Q. LncRNA BACE1-AS Promotes Autophagy-Mediated Neuronal Damage Through The miR-214-3p/ATG5 Signalling Axis In Alzheimer's Disease. Neuroscience 2021, 455, 52-64, doi:10.1016/j.neuroscience.2020.10.028.
- Zhang, X.; Song, W. The role of APP and BACE1 trafficking in APP processing and amyloid-beta generation. Alzheimers Res Ther 2013, 5, 46, doi:10.1186/alzrt211.
- Yan, R.; Han, P.; Miao, H.; Greengard, P.; Xu, H. The transmembrane domain of the Alzheimer's beta-secretase (BACE1) determines its late Golgi localization and access to beta -amyloid precursor protein (APP) substrate. J Biol Chem 2001, 276, 36788-36796, doi:10.1074/jbc.M104350200.
- Hansson Petersen, C.A.; Alikhani, N.; Behbahani, H.; Wiehager, B.; Pavlov, P.F.; Alafuzoff, I.; Leinonen, V.; Ito, A.; Winblad, B.; Glaser, E.; et al. The amyloid beta-peptide is imported into mitochondria via the TOM import machinery and localized to mitochondrial cristae. Proc Natl Acad Sci U S A 2008, 105, 13145-13150, doi:10.1073/pnas.0806192105.
- Greenfield, J.P.; Tsai, J.; Gouras, G.K.; Hai, B.; Thinakaran, G.; Checler, F.; Sisodia, S.S.; Greengard, P.; Xu, H. Endoplasmic reticulum and trans-Golgi network generate distinct populations of Alzheimer beta-amyloid peptides. Proc Natl Acad Sci U S A 1999, 96, 742-747, doi:10.1073/pnas.96.2.742.
- Cook, D.G.; Forman, M.S.; Sung, J.C.; Leight, S.; Kolson, D.L.; Iwatsubo, T.; Lee, V.M.; Doms, R.W. Alzheimer's A beta(1-42) is generated in the endoplasmic reticulum/intermediate compartment of NT2N cells. Nat Med 1997, 3, 1021-1023, doi:10.1038/nm0997-1021.
- Hartmann, T.; Bieger, S.C.; Bruhl, B.; Tienari, P.J.; Ida, N.; Allsop, D.; Roberts, G.W.; Masters, C.L.; Dotti, C.G.; Unsicker, K.; et al. Distinct sites of intracellular production for Alzheimer's disease A beta40/42 amyloid peptides. Nat Med 1997, 3, 1016-1020, doi:10.1038/nm0997-1016.
- Xu, H.; Sweeney, D.; Wang, R.; Thinakaran, G.; Lo, A.C.; Sisodia, S.S.; Greengard, P.; Gandy, S. Generation of Alzheimer beta-amyloid protein in the trans-Golgi network in the apparent absence of vesicle formation. Proc Natl Acad Sci U S A 1997, 94, 3748-3752, doi:10.1073/pnas.94.8.3748.
- Cataldo, A.M.; Petanceska, S.; Terio, N.B.; Peterhoff, C.M.; Durham, R.; Mercken, M.; Mehta, P.D.; Buxbaum, J.; Haroutunian, V.; Nixon, R.A. Abeta localization in abnormal endosomes: association with earliest Abeta elevations in AD and Down syndrome. Neurobiol Aging 2004, 25, 1263-1272, doi:10.1016/j.neurobiolaging.2004.02.027.
- Pasternak, S.H.; Callahan, J.W.; Mahuran, D.J. The role of the endosomal/lysosomal system in amyloid-beta production and the pathophysiology of Alzheimer's disease: reexamining the spatial paradox from a lysosomal perspective. J Alzheimers Dis 2004, 6, 53-65, doi:10.3233/jad-2004-6107.
- Takahashi, R.H.; Almeida, C.G.; Kearney, P.F.; Yu, F.; Lin, M.T.; Milner, T.A.; Gouras, G.K. Oligomerization of Alzheimer's beta-amyloid within processes and synapses of cultured neurons and brain. J Neurosci 2004, 24, 3592-3599, doi:10.1523/JNEUROSCI.5167-03.2004.
- Lee, E.K.; Park, Y.W.; Shin, D.Y.; Mook-Jung, I.; Yoo, Y.J. Cytosolic amyloid-beta peptide 42 escaping from degradation induces cell death. Biochem Biophys Res Commun 2006, 344, 471-477, doi:10.1016/j.bbrc.2006.03.166.
- Buckig, A.; Tikkanen, R.; Herzog, V.; Schmitz, A. Cytosolic and nuclear aggregation of the amyloid beta-peptide following its expression in the endoplasmic reticulum. Histochem Cell Biol 2002, 118, 353-360, doi:10.1007/s00418-002-0459-2.
- Snezhkina, A.V.; Kudryavtseva, A.V.; Kardymon, O.L.; Savvateeva, M.V.; Melnikova, N.V.; Krasnov, G.S.; Dmitriev, A.A. ROS Generation and Antioxidant Defense Systems in Normal and Malignant Cells. Oxid Med Cell Longev 2019, 2019, 6175804, doi:10.1155/2019/6175804.
- Dumont, M.; Ho, D.J.; Calingasan, N.Y.; Xu, H.; Gibson, G.; Beal, M.F. Mitochondrial dihydrolipoyl succinyltransferase deficiency accelerates amyloid pathology and memory deficit in a transgenic mouse model of amyloid deposition. Free Radic Biol Med 2009, 47, 1019-1027, doi:10.1016/j.freeradbiomed.2009.07.008.
- Shi, Q.; Xu, H.; Yu, H.; Zhang, N.; Ye, Y.; Estevez, A.G.; Deng, H.; Gibson, G.E. Inactivation and reactivation of the mitochondrial alpha-ketoglutarate dehydrogenase complex. J Biol Chem 2011, 286, 17640-17648, doi:10.1074/jbc.M110.203018.
- Karakas, D.; Ozpolat, B. The Role of LncRNAs in Translation. Noncoding RNA 2021, 7, doi:10.3390/ncrna7010016.
- Zeng, T.; Ni, H.; Yu, Y.; Zhang, M.; Wu, M.; Wang, Q.; Wang, L.; Xu, S.; Xu, Z.; Xu, C.; et al. BACE1-AS prevents BACE1 mRNA degradation through the sequestration of BACE1-targeting miRNAs. J Chem Neuroanat 2019, 98, 87-96, doi:10.1016/j.jchemneu.2019.04.001.
- Rangan, P.; Lobo, F.; Parrella, E.; Rochette, N.; Morselli, M.; Stephen, T.L.; Cremonini, A.L.; Tagliafico, L.; Persia, A.; Caffa, I.; et al. Fasting-mimicking diet cycles reduce neuroinflammation to attenuate cognitive decline in Alzheimer's models. Cell Rep 2022, 40, 111417, doi:10.1016/j.celrep.2022.111417.
- Liu, Y.; Hu, P.P.; Zhai, S.; Feng, W.X.; Zhang, R.; Li, Q.; Marshall, C.; Xiao, M.; Wu, T. Aquaporin 4 deficiency eliminates the beneficial effects of voluntary exercise in a mouse model of Alzheimer's disease. Neural Regen Res 2022, 17, 2079-2088, doi:10.4103/1673-5374.335169.
- Grochowska, K.M.; Yuanxiang, P.; Bar, J.; Raman, R.; Brugal, G.; Sahu, G.; Schweizer, M.; Bikbaev, A.; Schilling, S.; Demuth, H.U.; et al. Posttranslational modification impact on the mechanism by which amyloid-beta induces synaptic dysfunction. EMBO Rep 2017, 18, 962-981, doi:10.15252/embr.201643519.
- Liu, X.; Zhou, Q.; Zhang, J.H.; Wang, K.Y.; Saito, T.; Saido, T.C.; Wang, X.; Gao, X.; Azuma, K. Microglia-Based Sex-Biased Neuropathology in Early-Stage Alzheimer's Disease Model Mice and the Potential Pharmacologic Efficacy of Dioscin. Cells 2021, 10, doi:10.3390/cells10113261.
- Lee, S.H.; Chen, Y.H.; Chien, C.C.; Yan, Y.H.; Chen, H.C.; Chuang, H.C.; Hsieh, H.I.; Cho, K.H.; Kuo, L.W.; Chou, C.C.; et al. Three month inhalation exposure to low-level PM2.5 induced brain toxicity in an Alzheimer's disease mouse model. PLoS One 2021, 16, e0254587, doi:10.1371/journal.pone.0254587.
- Niszczota, P.; Gieras, M. Photographic data on the influence of the composition, preparation method, time and fuel system on the size of water droplets in a fuel-water emulsion. Data Brief 2022, 43, 108406, doi:10.1016/j.dib.2022.108406.
- Song, X.; Sun, Y.; Wang, Z.; Su, Y.; Wang, Y.; Wang, X. Exendin-4 alleviates beta-Amyloid peptide toxicity via DAF-16 in a Caenorhabditis elegans model of Alzheimer's disease. Front Aging Neurosci 2022, 14, 955113, doi:10.3389/fnagi.2022.955113.
- Serebrovska, Z.O.; Xi, L.; Tumanovska, L.V.; Shysh, A.M.; Goncharov, S.V.; Khetsuriani, M.; Kozak, T.O.; Pashevin, D.A.; Dosenko, V.E.; Virko, S.V.; et al. Response of Circulating Inflammatory Markers to Intermittent Hypoxia-Hyperoxia Training in Healthy Elderly People and Patients with Mild Cognitive Impairment. Life (Basel) 2022, 12, doi:10.3390/life12030432.
- Lyu, Z.; Chan, Y.; Li, Q.; Zhang, Q.; Liu, K.; Xiang, J.; Li, X.; Cai, D.; Li, Y.; Wang, B.; et al. Destructive Effects of Pyroptosis on Homeostasis of Neuron Survival Associated with the Dysfunctional BBB-Glymphatic System and Amyloid-Beta Accumulation after Cerebral Ischemia/Reperfusion in Rats. Neural Plast 2021, 2021, 4504363, doi:10.1155/2021/4504363.
- Stroud, J.C.; Liu, C.; Teng, P.K.; Eisenberg, D. Toxic fibrillar oligomers of amyloid-beta have cross-beta structure. Proc Natl Acad Sci U S A 2012, 109, 7717-7722, doi:10.1073/pnas.1203193109.
- Nagele, R.G.; D'Andrea, M.R.; Lee, H.; Venkataraman, V.; Wang, H.Y. Astrocytes accumulate A beta 42 and give rise to astrocytic amyloid plaques in Alzheimer disease brains. Brain Res 2003, 971, 197-209, doi:10.1016/s0006-8993(03)02361-8.
- Gu, L.; Guo, Z. Alzheimer's Abeta42 and Abeta40 peptides form interlaced amyloid fibrils. J Neurochem 2013, 126, 305-311, doi:10.1111/jnc.12202.
- Gabr, A.A.; Lee, H.J.; Onphachanh, X.; Jung, Y.H.; Kim, J.S.; Chae, C.W.; Han, H.J. Ethanol-induced PGE(2) up-regulates Abeta production through PKA/CREB signaling pathway. Biochim Biophys Acta Mol Basis Dis 2017, 1863, 2942-2953, doi:10.1016/j.bbadis.2017.06.020.
- Liang, Y.; Harris, F.L.; Brown, L.A. Alcohol induced mitochondrial oxidative stress and alveolar macrophage dysfunction. Biomed Res Int 2014, 2014, 371593, doi:10.1155/2014/371593.
- Kumar, A.; Davuluri, G.; Welch, N.; Kim, A.; Gangadhariah, M.; Allawy, A.; Priyadarshini, A.; McMullen, M.R.; Sandlers, Y.; Willard, B.; et al. Oxidative stress mediates ethanol-induced skeletal muscle mitochondrial dysfunction and dysregulated protein synthesis and autophagy. Free Radic Biol Med 2019, 145, 284-299, doi:10.1016/j.freeradbiomed.2019.09.031.
- Fernandez-Checa, J.C.; Kaplowitz, N.; Colell, A.; Garcia-Ruiz, C. Oxidative stress and alcoholic liver disease. Alcohol Health Res World 1997, 21, 321-324.
- Ohyagi, Y.; Asahara, H.; Chui, D.H.; Tsuruta, Y.; Sakae, N.; Miyoshi, K.; Yamada, T.; Kikuchi, H.; Taniwaki, T.; Murai, H.; et al. Intracellular Abeta42 activates p53 promoter: a pathway to neurodegeneration in Alzheimer's disease. FASEB J 2005, 19, 255-257, doi:10.1096/fj.04-2637fje.
- von Mikecz, A. Pathology and function of nuclear amyloid. Protein homeostasis matters. Nucleus 2014, 5, 311-317, doi:10.4161/nucl.29404.

Reviewer 2 Report
Comments and Suggestions for Authors
The research conducted by Kumar et al. investigates the connection between alcohol-use-disorder and Alzheimer’s-like neurological disorders. The study discovered that when astrocytes are exposed to ethanol, there is an increase in markers of Alzheimer’s Disease such as amyloid precursor protein, Aβ 1-42, β-site cleaving enzyme, oxidative stress marker-4HNE, proinflammatory cytokines (TNF-α, IL1β, IL6), lncRNA BACE1-AS, and alcohol metabolizing enzymes. The results are intriguing and well-articulated, particularly in the discussion section. However, the abstract could be more concise. For instance, the sentence, “The role of astrocytes as contributors to Alzheimer’s-like pathology in HIV-associated neurological disorder as well as in opiate-use-disorder, has been previously reported by our group,” seems unnecessary in the abstract.
Major point.
1. In the original blot, the few antibodies seem to bind nonspecifically but strongly at other molecular weights. To confirm these were the correct bands, the author should perform an immunoprecipitation with these antibodies for at least one concentration.
Minor corrections:
2. In the introduction, on line 85, you mention the downregulation of neuroprotective genes. Could you provide the names of a few genes that were relevant to this study?
3. In Fig.2 A and B, the western blot image for the control and 6.25mM appears similar, but the quantification looks twice that of the control in 6.2mM. Could you please verify this?
4. Could you clarify in the figure legend what n=6/group refers to? Does it mean 6 cells?
5. For the control at 96mM, in Fig 3H, 4A and D, the western blot image doesn’t match with the quantification. Could you please look into this?
Author Response
We are thankful to the editor and reviewer for the constructive comments. For better visibility of the individual replicates, we have edited all the figures. We have addressed each of the comments and highlighted the changes in yellow in the main text.
Reviewer 2:
The research conducted by Kumar et al. investigates the connection between alcohol-use-disorder and Alzheimer’s-like neurological disorders. The study discovered that when astrocytes are exposed to ethanol, there is an increase in markers of Alzheimer’s Disease such as amyloid precursor protein, Aβ 1-42, β-site cleaving enzyme, oxidative stress marker-4HNE, proinflammatory cytokines (TNF-α, IL1β, IL6), lncRNA BACE1-AS, and alcohol metabolizing enzymes. The results are intriguing and well-articulated, particularly in the discussion section. However, the abstract could be more concise.
For instance, the sentence, “The role of astrocytes as contributors to Alzheimer’s-like pathology in HIV-associated neurological disorder as well as in opiate-use-disorder, has been previously reported by our group,” seems unnecessary in the abstract.
Response: We are thankful for the reviewer's suggestion. Appropriate changes have been made in the abstract.
Major point
- In the original blot, the few antibodies seem to bind nonspecifically but strongly at other molecular weights. To confirm these were the correct bands, the author should perform an immunoprecipitation with these antibodies for at least one concentration.
Response: Immunoprecipitation (IP) with AβmOC64 & IgG antibodies has been performed, and results demonstrate, Aβ specific bands (37-40 KDa), the other 2- bands at 50 KDa (heavy chain) and ~20 KDa (light chain), are probably due to IgG. Additionally, in the rightmost lane, the Alcohol exposed (12.5 mM) astrocyte samples have been loaded (labeled as positive control) shows a clear band at 37-40 kDa. The IP results are enclosed herewith:
Note: For BCAE1: We have replaced the western blot data with new results showing a single band around ~56 kDa.
For CYP2E1: Due to the unavailability of an IP-specific antibody, we were unable to perform the IP. However, we would like to mention that the band in the original blot was identified based on the results shown on the Abcam website, indicating a band around ~56 kDa. (Anti-Cytochrome P450 2E1 antibody (ab28146) | Abcam).
Minor corrections:
- In the introduction, on line 85, you mention the downregulation of neuroprotective genes. Could you provide the names of a few genes that were relevant to this study?
Response: In the corresponding study the authors have demonstrated that ethanol downregulates genes such as APOE, TREM2, Lipoprotein lipase (LPL), and Cathepsin D (CTSD). However, all of these genes are not a focus of this study.
- In Fig. 2A and B, the western blot image for the control and 6.25mM appears similar, but the quantification looks twice that of the control in 6.2mM. Could you please verify this?
Response: New quantifications have been included in Fig. 2A and B. (Page no. 7, Line no. 244-245)
- Could you clarify in the figure legend what n=6/group refers to? Does it mean 6 cells?
Response: n=6/ group here represents six biological replicates (six passages of human astrocytes from different donors) for each concentration of ethanol. Additional details have been made in the figure legend.
- For the control at 96mM, in Fig 3H, 4A and D, the western blot image doesn’t match with the quantification. Could you please look into this?
Response: For Figure 3H, new quantification has been added. Regarding Figure 4A, our quantification provides the same result as already shown in the figure. Upon closer inspection, the 96 mM band on the western blot is nearly double the intensity of the control band, which is evident in the graph's quantification. Additionally, the actin band for the 96 mM is also less intense than the control actin band. Therefore, upon normalization, the ADH band with the corresponding actin band gives a quantification almost twice that of the control level. Furthermore, for Figure 4D, the 4-HNE band appears significantly more intense compared to the control. However, it is also noticeable that the actin band is very intense for the 96 mM concentration. Therefore, normalizing the 96 mM 4-HNE to the corresponding actin reduces the mean values.

Round 2
Reviewer 1 Report
Comments and Suggestions for Authors
The authors have thoroughly addressed all major queries. Fig.7 looks more complete. The manuscript can be accepted in its present form.